# Developing Graph Convolutional Networks and Mutual Information for Arrhythmic Diagnosis Based on Multichannel ECG Signals

**DOI:** 10.3390/ijerph191710707

**Published:** 2022-08-28

**Authors:** Bahare Andayeshgar, Fardin Abdali-Mohammadi, Majid Sepahvand, Alireza Daneshkhah, Afshin Almasi, Nader Salari

**Affiliations:** 1Department of Biostatistics, School of Health, Kermanshah University of Medical Sciences, Kermanshah 6715847141, Iran; 2Department of Computer Engineering and Information Technology, Razi University, Kermanshah 6714967346, Iran; 3Research Centre for Computational Science and Mathematical Modelling, Coventry University, Coventry CV1 2JH, UK; 4Clinical Research Development Center, Imam Khomeini and Mohammad Kermanshahi and Farabi Hospitals, Kermanshah University of Medical Sciences, Kermanshah 6715847141, Iran; 5Sleep Disorders Research Center, Kermanshah University of Medical Sciences, Kermanshah 6715847141, Iran

**Keywords:** heart arrhythmia types, ECG-based diagnostic, graph convolutional networks, CNN, mutual information

## Abstract

Cardiovascular diseases, like arrhythmia, as the leading causes of death in the world, can be automatically diagnosed using an electrocardiogram (ECG). The ECG-based diagnostic has notably resulted in reducing human errors. The main aim of this study is to increase the accuracy of arrhythmia diagnosis and classify various types of arrhythmias in individuals (suffering from cardiovascular diseases) using a novel graph convolutional network (GCN) benefitting from mutual information (MI) indices extracted from the ECG leads. In this research, for the first time, the relationships of 12 ECG leads measured using MI as an adjacency matrix were illustrated by the developed GCN and included in the ECG-based diagnostic method. Cross-validation methods were applied to select both training and testing groups. The proposed methodology was validated in practice by applying it to the large ECG database, recently published by Chapman University. The GCN-MI structure with 15 layers was selected as the best model for the selected database, which illustrates a very high accuracy in classifying different types of rhythms. The classification indicators of sensitivity, precision, specificity, and accuracy for classifying heart rhythm type, using GCN-MI, were computed as 98.45%, 97.89%, 99.85%, and 99.71%, respectively. The results of the present study and its comparison with other studies showed that considering the MI index to measure the relationship between cardiac leads has led to the improvement of GCN performance for detecting and classifying the type of arrhythmias, in comparison to the existing methods. For example, the above classification indicators for the GCN with the identity adjacency matrix (or GCN-Id) were reported to be 68.24%, 72.83%, 95.24%, and 92.68%, respectively.

## 1. Introduction

Cardiovascular diseases are one of the main causes of mortality across the world, which impose a heavy burden on society [1,2,3]. Cardiac arrhythmia, which is known as an abnormal heart rhythm, is considered as a very important category of cardiovascular disorders [4]. There are different types of arrhythmias with different symptoms such as a heartbeat that is too slow or fast (sinus bradycardia (SB) and atrial tachycardia (AT)), and irregular rhythms with missing or distorted sections and intervals (premature ventricular contraction (AVC)). Atrial fibrillation (AFIB), as the most common and dangerous type of arrhythmia, is associated with a significant increase in the risk of severe heart failure and stroke [5]. Other types of arrhythmias include Sinus Tachycardia (ST), Sinus Irregularity (SI), Supraventricular Tachycardia (ST), Atrioventricular Node Reentrant Tachycardia (AVNRT), Atrioventricular Reentrant Tachycardia (AVRT), and Sinus Atrium to Atrial Wandering Rhythm (SAAWR).

Diagnostic methods of arrhythmia include electrocardiograms, Holter and event monitors, implantable loop recorders, stress tests, echocardiograms (ECGs), and angiography. Despite the development of many advanced non-invasive measurement technologies in the last 100 years, the 12-lead ECG has still been widely used worldwide and become established as the de facto standard for the non-invasive assessment of a wide range of heart diseases [1]. Accessibility and affordability are the other merits of non-invasive ECGs [6]. In the 12-lead ECG, 10 electrodes are placed on the surface of the patient’s chest. The total electrical potential of the heart is then measured from 12 different points and recorded over a period (usually 10 s) [7].

The type of arrhythmia is diagnosed according to the waveform of the electrocardiogram. Correct and quick diagnosis of cardiac arrhythmias is very important for the prevention and treatment of heart diseases [6]. Misinterpretations of ECGs may lead to incorrect clinical decisions and adverse outcomes [8]. Interpretation by ECG requires sufficient expertise and training, and the observer’s knowledge directly influences the accuracy, so applying automatic diagnosis of arrhythmias using ECG signals, can be useful in preventing human errors [3,9,10]. Recently, the use of neural network and deep learning techniques, to analyze and extract features of the ECG data, has received significant attention [11,12,13].

Traditional machine learning schemes are heavily influenced by data, which requires complex preprocessing such as noise cancellation and data normalization. Additionally, they are vulnerable to over-extraction of unnecessary features, filter design, and additional trait sorting, and eventually introduce another classification algorithm [14]. A type of neural network that partially alleviates these problems is the convolutional neural network (CNN), which automatically detects important features without human supervision [15]. The convolutional neural network (CNN) is a class of artificial neural networks that has widely been used in various computer vision and machine learning applications and has received considerable attraction across a variety of medical fields, including radiology, cancer pathology, diagnosis of cardiovascular disease, etc. The CNN is constructed to learn spatial hierarchies of features automatically and adaptively [15,16] through backpropagation by using multiple building blocks, such as convolution layers, pooling layers, and fully connected layers. The CNN is also a useful tool for solving the problem of pattern recognition [17], which has wide applications such as image, sound, and signal analysis, etc.

There are several studies that used convolutional networks with various architectures for diagnosing cardiac arrhythmias by deeply learning ECG data [4,18,19,20,21,22,23,24]. In each of these studies, the adjusted convolutional network plays an important role to increase the diagnosis accuracy. The size of data used in these studies was not adequate due to limited subject records and particularly the ECG beats extracted from the same individuals. This data scarcity would reduce the generalizability of the constructed models. Furthermore, since the ECG examines heart function from different angles (leads), these leads have common and useful information that can be measured and should be included in the developed deep learning methods reported in [4,18,19,20,21,22,23,24]. In other words, it is very important to consider the structure of 12-leads ECG data, or the relationship between the leads, to better train the CNNs and thus efficiently learn about the ECG data.

The main aim of this study is to learn the lead structures and use them to increase accuracy of the diagnosis and classification of cardiac arrhythmias. Therefore, this study proposed the GCN as an effective deep learning modeling tool, which enables us to include the lead structures to enhance the accuracy of arrhythmia diagnosis and classify various types of arrhythmias.

One of the strengths of the GCNs is their ability to learn complex structures, by performing convolution operations on graphs instead of pixel images [25], which make them very effective in modeling graph-structured data [26]. A graph, as a type of data structure, consists of nodes and connections which can be used when modeling complex real-world phenomena. Unlike the CNN which should use fixed square kernels when learning the data, GCNs can use correlations between adjacent nodes and perform a flexible convolution in randomly irregular areas [27].

In this study, the ECG data collected for each person are considered as a graph and the heart leads as nodes. The shared information between the common leads data is measured with the mutual information index (MI) and the connection weights are provided to the network to improve learning performance. For the data with a complex dependence structure, the Pearson correlation indices can be used since the electrical potential records of each lead are represented as a time series and their dependencies are measured using autocorrelation indices. However, the MI can measure both linear and nonlinear relationships between two time series and calculate the common information between them using the entropy function [28]. In this study, the MI index was used to measure the correlation between leads and then construct a graph between them, which is required to effectively learn the GCN. The GCN-MI-based model constructed in this study will be then used to diagnose and classify the type of arrhythmia in individuals suffering from cardiovascular diseases.

In brief, the main contributions of this study are:Representing an individual’s ECG information as a graph.Using the MI index to measure the relationship of leads and structure the graph.Proposing the GCN-MI, for the first time, to diagnose and classify the type of arrhythmia.Examining a very recent ECG dataset with a considerably large size (at least 10,000 times larger than the datasets used in [4,18,19,20,21,22,23,24]).

## 2. Materials and Methods

### 2.1. Data

The study used a newly opened research database for 12-lead electrocardiogram signals under the supervision of Chapman University and the Public Hospital (Medical School of Zhejiang University in Shaoxing Hospital). The database’s aim is to enable the scientific community to conduct new research on arrhythmias and other cardiovascular diseases, and it contains 12 ECGs from 10,646 patients at a 500 Hz sampling rate with 11 normal rhythms and 67 additional cardiovascular diseases. The ECG samples are labeled by professionals and taken within 10 s for each person, where each ECG sample contains 5000 rows and 12 columns.

Statistically speaking, the number of samples is equal to 10,646 people whose ECG results are measured as 10,646 12-lead variable time series with 5000 points. In each variable (lead), the intensity of the electric current is recorded in 5000 turns (every 10 s).

The data included 5956 men and 4690 women, of which 17% had a normal sinus rhythm and 83% had at least one abnormality (Table 1) [5]. The relative frequency of rhythm types is shown in this dataset (Figure 1). Figure 2 shows the frequency of each rhythm by gender.

### 2.2. Graph Convolutional Network

#### 2.2.1. Introduction to the GCN

Graph neural network was introduced in 2009, which is based on graph theory and forms the basis of various types of graph networks. GCNs, as one of the most popular graph networks, mainly use the combination of the Fourier transform and the Taylor expansion formula to improve filter performance, and with their excellent performance have been widely used in the classification of diseases [29]. GCNs are a group of deep neural networks that perform mathematical operations called convolution on graphs, and their purpose is to categorize graphs, nodes, or connections [30].

The graph can be represented by $G=\left(V,\;A\right),\;V\in R_{N\times f}\;$ is the matrix of the vertex signal which displays N nodes that each have ƒ features. $A\in R_{N\times N}\;$ indicates the adjacency matrix that shows edge information and the relationship between nodes. If there is an edge between node i and node  j, A (i, j) indicates the weight of that edge i=1, 2… N, and  j=1, 2… N, otherwise A (i, j)=0.

#### 2.2.2. Convolution Graph

Graph data can briefly display information on edges and vertices, in which, in order to process and learn this information, a convolutional filtering method must be considered to filter both edge and vertex information. This is a spatial approach to the graph convolution method that follows the local neighborhood graph filtering strategy.

The convolution function of a graph is based on matrix polynomials in proximity to the graph (see also Zhang et al., 2019 [25], for further details).


(1)
$$H=h_{0}I+h_{1}A^{1}+h_{2}A^{2}+h_{3}A^{3}+\ldots +h_{k}A^{k}$$


This filter is defined as polynomials of the degree k of the graph adjacency matrix. A is the proximity matrix and its power represents the number of steps of a given vertex multiplied by the assumed filter coefficients. Scalar coefficients $h_{i}$ control the participation of neighbors of one vertex during convolution operation, so the filter matrix is obtained as H ∈ ℝ_*N*__×__*N*_. The convolution of vertices V with filter H, defined in (1), is a matrix multiplication shown below.

$$V_{out}=HV_{in}$$where $V_{out},\;V_{in}\in \;$ℝ_*N*_, $V_{in}$ is the input vertex matrix and $V_{out}$ is the vertex matrix after the filter operation. Further details about how this model can be adjusted and how the convolution filter of the selected graph (i.e., H) can be computed are provided in [App app-ijerph-19-10707]. 

#### 2.2.3. The Architecture for the Proposed GCN 

The first step is preparing the input graph in a way that active and inactive nodes are equated and considered equal. Additionally, a separate entry will be considered as a label next to it, and the output of this step will be entered as the graph $z_{0}$ in the next step.

In the second step, convolution operation will be performed on this graph in several layers, in a way so that the desired filter will be applied on each node according to its neighboring edge (Figure 3). This architecture shows a GCN with two layers of convection and graphs with 12 nodes. In the convolution layers, this position is indicated by red nodes at each stage. The output of the first layer of convolution $z_{1}$ is given to the second layer of the convolution, and $z_{2}$ is generated as the output of the second layer.

In the third step, a decision is made about the response variable by linearizing the created properties and delivering them to a Softmax layer. To do this, first by putting together all the created graphs from their different permutations from $z_{0}$ to $z_{2}$, several linear vectors will be generated, and each of these vectors is called a property vector. Figure 3 displays the two steps of the concatenate operation in the third step. The output of these two stages will be a linear vector that will enter the fully connected neural network at the last level. Eventually, using this network, known as Softmax, the decision boundary will be then created. 

### 2.3. Mutual Information

The ECG examines the heart function from different angles (leads), and these leads have common information (communication) that can be measured. This study measured this relationship using the MI index and introduced it to the GCN under the adjacency matrix.

In the following, each lead is assumed as a variable, and the method of calculating MI between two variables (X, Y) is explained.

Mutual information is a measure of the dependence between two variables which is the amount of information obtained by observing Y from X. MI comes from the definition of entropy. In the discrete mode, the mutual information of two variables X and Y is obtained as follows.


$$\text{I}\left(\text{X};\;Y\right)\;=\underset{\text{x}\in \text{X},\text{y}\in \text{Y}}{\sum }\;\mathrm{Pr}\left[X\;=\;x,Y\;=\;y\right]\cdot \;\log\;\left(\frac{\mathrm{Pr}\left[X\;=\;x,Y\;=\;y\right]}{\mathrm{Pr}\left[X\;=\;x\right]\cdot \mathrm{Pr}\left[Y\;=\;y\right]}\right)$$


This can be seen as the Kullback–Leibler divergence measure between the joint distribution Pr[X=x, Y=y] and the product distribution Pr[X=x]. Pr[Y=y]. As a result, the mutual information can similarly be expressed as the expected value, over X, of the divergence between the conditional probability Pr[Y=y| X=x] and the marginal probability Pr[Y=y]:$$I\left(X;\;Y\right)=\underset{x\in X}{\sum }\mathrm{Pr}\left[X\;=\;x\right]\underset{y\in Y}{\sum }\mathrm{Pr}\left[Y\;=\;y|X\;=\;x\right]\;\cdot \log\left(\frac{\mathrm{Pr}\left[Y\;=\;y|X\;=\;x\right]}{\mathrm{Pr}\left[Y\;=\;y\right]}\right)$$

The mutual information is always greater than or equal to zero. In fact, if X and Y are independent, it becomes zero. 

For the case that both variables X and Y are continuous, the corresponding relationships can be expressed in terms of integrals [31,32,33]. These relationships are computed and stored in the ${\mathrm{MI}}_{N\times N}$ matrix (where, N is equal to 12) for 12 ECG leads data considered in this study, using R software (R Foundation for Statistical Computing, Vienna, Austria). 

### 2.4. Methodology

This study aimed to diagnose and classify the type of heart rhythm using a GCN with an MI proximity matrix. For this purpose, the newly opened research database of Chapman University, which has electrocardiogram signals from more than 10,000 people, was used. 

The steps required to conduct the proposed method for ECG analysis and classify heart rhythm type are: (1) data collection; (2) data pre-processing; (3) model building, training, and tuning; and finally, validation and classification. 

The data used in this paper were originally collected and introduced by Zheng et al. [5]. As discussed in Section 2.1, in the data collection phase, first, each subject was exposed to a 12-lead resting ECG for 10 s. The collected data were stored in a GE MUSE ECG system. Then, a licensed physician labeled the rhythms. Another licensed physician performed secondary validation. If there was a dispute, a senior physician intervened and made the final decision. The ECG data and diagnostic information from the GE MUSE system were then transferred into XML segments and encoded with a specific nomenclature defined by General Electric (GE). Finally, a conversion tool was used to extract ECG data and diagnostic information from the XML file and transfer them into CSV format [5].

In the second stage (pre-processing), the noise reduction operation was carried out. The sources of noise contamination in the ECG data, as reported in [5] and used in this study, would include power line interference, motion artifacts, electrode contact noise, baseline wandering, muscle contraction, and random noise. In order to statistically analyze the ECG data, the noise should be removed from the data. 

Zhang et al. [5] proposed and implemented a sequential denoising approach for removing the noise from the raw ECG data. Since the normal range of the ECG frequency is from 0.5 to 50 Hz, a Butterworth low pass filter was used to remove the signal above 50 Hz. In order to clear the effects of baseline wandering, a LOESS smoother was then utilized. Finally, the non-local means technique was used to handle the remaining noise (see also “§. Data denoising method” and Figure 3, Figure 4 and Figure 5 presented in [5], for the details of data pre-processing steps for the dataset analyzed in this study). 

The third step, which would be one the main contributions of this study, is to construct or learn a graph for each ECG by calculating the MI matrix and setting up and training the GCN network, as demonstrated in Figure 4. The details of this step are discussed below.

In order to construct the graph, a graph was first shown to each expert to show heart leads and 12 nodes represent 12 heart leads. Considering that each lead contains 5000 samples of the heart’s electrical potential, the feature matrix measures 12 × 5000. The relationship of the leads was calculated with the MI matrix and used as the adjacency matrix (12 × 12) in the GCN (Figure 4).

There are 11 rhythms in the data set, but the number of samples for 4 types of these rhythms is less than 2%, AT (n = 121; 1.14%), AVNRT (n = 16; 0.15%), AVRT (n = 8; 0.07%), and SAA (n = 7; 0.07%). Therefore, information about the other 7 rhythms was used for training and testing (10,494 people). These 7 rhythms included AF, AFIB, SI, SB, SR, ST, and SVT. In the first stage, the appropriate GCN-MI was selected, and this model was trained with 5, 10, and 15 layers. For this purpose, cross-validation methods with k = 2, 3, 4, 5-fold and leave-one-out cross-validation methods were used. These CV methods were also used to select the training and testing groups of the GCN-MI networks with 5, 10, and 15 different layers. The performance of these networks was assessed using standard evaluation criteria, such as accuracy, specificity, precision, and sensitivity, in the testing set. To select the most appropriate accuracy network, three networks were compared in pairs by t-independent test. A network was then selected, and all data were categorized by it. To evaluate this classification, a confusion matrix was prepared, and evaluation criteria were estimated. 

Finally, in order to show the novelty of this article in improving the performance of the GCN-MI network in the diagnosis of arrhythmias and classifying different types of rhythms, the results were compared with those in which no structure between cardiac leads was considered. In the unstructured state, the identity matrix was considered as the adjacency matrix. The results of this study were also compared with other studies that have used different methods to diagnose arrhythmias in this dataset.

Considering true positive (TP), true negative (TN), false positive (FP), and false negative (FN), the accuracy, sensitivity, specificity, and precision, as performance criteria, of the proposed CGN-based classification method, are defined as follows:


$$\mathrm{Accuracy}=\frac{\mathrm{TP}+\mathrm{TN}}{\mathrm{TP}+\mathrm{TN}+\mathrm{FP}+\mathrm{FN}}$$



$$\mathrm{Sensitivity}=\frac{\mathrm{TP}}{\mathrm{TP}+\mathrm{FN}}$$



$$\mathrm{Specificity}=\frac{\mathrm{TN}}{\mathrm{FP}+\mathrm{TN}}$$



$$\mathrm{Precision}=\frac{\mathrm{TP}}{\mathrm{TP}+\mathrm{FP}}$$


## 3. Experimental Results

The MI matrix was calculated using R software to measure the relationship of 12 heart leads for all individuals and was introduced as a proximity matrix in the GCN using Python software. GCN-MI-10 and GCN-MI-15 were formed to find an optimal network of GCN-MI-5 (convolution layers) and their performance was also evaluated under leave-one-out cross-validation with k = 2, 3, 4, 5. Accuracy, specificity, sensitivity, and precision indices were used to assess the performance of these networks in a test-set with cross-validation methods (Table 2). From Figure 5, it can be observed that the graph of GCN-MI-15 evaluation indicators is always higher than the other two networks. However, the t-independent test was used to assess the significance of this difference. The results, presented in Table 2 and Figure 5, showed that the accuracy of the GCN-MI-15 network for leave-one-out and for the cross-validation method with k = 2, 3, 4, 5 is significantly higher than the accuracy of GCN-MI-5 and GCN-MI-10 networks.

A statistical test was used to select the appropriate number of folds for cross-validation. The result of the t-independent test with 95% confidence showed that the accuracy of fold = 3 and 4 was higher than other selections.

However, it is obvious that adding more layers will help us to extract more features, but we should be aware of the possible overfitting by adding more layers and computational layers. Based on the above benchmark study between the accuracy and other predictive performance values of the GCN-MI with different layers, and negligible increased computational cost, the GCN-MI with 15 intermediate layers was thus selected and configured using cross-validation method and with 4 folds. In the selected spatial GCN model, for each convolution layer, as explained in Section 2.2.2, the convolution operation is performed based on matrix polynomials in proximity to the graph. The output of each layer of the convolution is given to the next layer. The parameters of the selected GCN network with 15 convolution layers and the MI adjacency matrix are shown in Table 3.

The function of the middle layers of this network was ReLU and the subordinate of the last layer was Softmax. The learning curve shows that the training accuracy is at a high level and the model fits are good. It also shows that after 600 epochs, the cross-validation accuracy converged to the training accuracy (Figure 6).

In the next step, using the tuned network, the type of rhythm of all individuals was identified and classified. The results showed that 100% of people classified by the selected network in SB were correctly identified. The results showed that the accuracy of the GCN-MI-15 network in the testing set with the cross-validation method was higher than other networks. Therefore, in the end, all data were classified by the GCN-MI network. The results showed that 99.6%, 98.1%, 99.5%, 95.5%, 92.5%, and 100% of people who were classified in SR, AFIB, ST, AF, SI, and SVT, respectively, were correctly diagnosed, and for the rest another type of arrhythmia was given. The confusion matrix shows how this classification works. The network with the identity matrix gave poor results (Figure 7).

In SB detection, this network had sensitivity, precision, specificity, and accuracy equal to 99.35%, 100%, 100%, and 99.76%, respectively, and these values for SR were equal to 98.79%, 99.61%, 99.92%, and 99.72%. The classification performance criteria result of other classes can be seen in Table 4. The results show that the sensitivity, precision, specificity, and accuracy of the GCN network with the proposed structure (MI) were considerably higher than the GCN network with the identity structure. The results also show that the overall values of sensitivity, precision, specificity, and accuracy of GCN-MI in diagnosing the type of arrhythmia are equal to 98.45%, 97.89%, 99.85%, and 99.71%, respectively. This network had the highest sensitivity (99.73%) for ST rhythm classification and the highest accuracy of 99.95% for SVT classification. Additionally, the highest precision and specificity were related to the classification of SB and SVT rhythms (100%). In contrast, the overall values of sensitivity, precision, specificity, and accuracy of GCN-Id are equal to 68.24%, 72.83%, 95.24%, and 92.68, respectively. The GCN-Id network had the highest sensitivity (90.21%) and precision (84.36%) for SB rhythm classification and the highest accuracy of 94.56 for SVT classification. Furthermore, the highest specificity was related to the classification of SI rhythms (98.05%) (Table 4).

An independent *t*-test was used to compare the accuracy of two GCN-MI models with the mean and standard deviation equal to 0.9971 ± 0.3210 and GCN-Id with the mean and standard deviation equal to 0.9268 ± 0.4774. The results of this test showed that the accuracy of the GCN-MI model in detecting and classifying the seven types of arrhythmias mentioned is significantly (*p*-value < 0.001) higher than the GCN-Id method.

The comparison of the results of the present study with other studies showed that the accuracy of the proposed method (99.71%) was higher than that of the DNN (92.24%), CNN Trees (97.60%), Meta CNN Trees (98.29%), Single Classifier (92.89%), 1-D CNN (94.01%), RNN (96.21%), XGBoost (89.40%), Teacher Model (98.96%), Student Model (98.13%), and GCN-MI (99.71%) models in arrhythmic diagnosis (Figure 8).

## 4. Discussion

This research aimed to distinguish and categorize different types of heart rhythms. To date, neither ECGs as a graph nor GCNs have been used directly to detect and classify cardiac arrhythmias, and this study is the first to consider the relationship of cardiac leads to the MI matrix and provide it to the network under the adjacency matrix. In this way, the new large ECG database of Chapman University was used. Our study recommended a 15-layer GCN-MI to detect and classify seven types of rhythms based on a 12-lead ECG involving more than 10,000 individuals. Comparing the evaluation indicators of the proposed network (GCN-MI) with GCN-Id showed that considering the MI index to measure the relationship between cardiac leads led to an improvement in GCN performance. The accuracy of the proposal model was 99.71%, while the accuracy of the GCN network with the identity structure was 92.68%. Many researchers, applying different sources and different approaches, have tried to create an arrhythmia detection system using a deep learning architecture. 

In a similar work conducted by Zheng Jiang et al., two CNN networks and then the GCN were used to diagnose multiple cardiac disorders. The developed GCN, in this work, is used to examine the relationship between arrhythmia classes when there is more than one cardiac disorder during ECG signal collection, and binary cross-entropy loss is used for correlation between labels. As it is evident, the inter-dependency of the heart leads is neglected in this study. This could be another strength of our proposed method where each ECG is defined as a graph and directly expressed by the GCN (with inter-dependency relationships between the leads determined using MI structure) to detect and classify arrhythmia type [34].

Shaker et al. also recommended the Generative Adversarial Networks to distinguish and classify rhythm types based on 15 different classes of MIT-BIH arrhythmic datasets, in which the overall accuracy was only 98.30% [35], in comparison to 99.71% of the proposed method in this paper.

Yao et al., in 2020, introduced an ATI-CNN to detect arrhythmic type based on a multi-channel ECG signal, which included 6877 12-lead ECG records, and classified a total of 8 types of arrhythmias. This study showed that ATI-CNN reached an overall accuracy of 81.2%. Comparing to a classic 16-layer CNN called VGG-net, the accuracy increased by an average of 7.7%, and in the detection of paroxysmal arrhythmias by 26.8% [22]. In 2020, Zhao and Ban proposed a way to combine a CNN and an extreme learning machine with the aim of automatically identifying and classifying ECG signals. Data from the MIT-BIH database was used in this study, which contained 48 ECG data, and resulted in an accuracy rate of 97.50% [24].

In 2019, Gao et al., in order to accurately detect the ECG, recommended the LSTM network with FL using the data in the MIT-BIH arrhythmia database and the accuracy was 99.26% [36]. Hannun et al., in 2019, developed the DNN to classify 12 rhythm classes applying 91,232 single-lead ECGs from 53,877 patients, with an AUC (below rock curve level) of 0.97 [37]. In 2019, Oh et al. suggested a modified U-net for automatically detecting and encoding five types of ECG arrhythmias. They used the MIT-BIH arrhythmia database, and the accuracy of their recommendation system was 97.32% [38]. Li et al., in 2019, introduced a residual network (ResNet) to identify five different types of heart rhythm using a 2-lead ECG, with an accuracy of 99.38%. The data were obtained from the MIT-BIH arrhythmia database [39]. Yildirim et al., in 2018, designed a 16-layer 1D-CNN to determine 17 classes of heart rhythm, using the MIT-BIH arrhythmia database, with a network accuracy of 91.33% [40].

In 2018, utilizing the ECGs of 47 individuals from the MIT-BIH arrhythmia database, Xu et al. suggested a DNN to classify five classes of arrhythmias, with an accuracy of 93.1% [41]. Acharya et al., in 2017, using a CNN network with nine layers, with an accuracy of 94.03%, categorized the arrhythmic type into five types of arrhythmias. The data were related to the MIT-BIH arrhythmia database [42].

Several studies have proposed models for the diagnosis of arrhythmias using Chapman data also used in the present study. The accuracy of DNN [43], CNN Trees [44], Meta CNN Trees [45], Single Classifier [46], 1-D CNN [47], RNN [48], XGBoost [49], Teacher and Student [50] models was 92.24%, 97.60%, 98.29%, 92.89%, 94.01%, 96.21%, 89.40%, 98.96%, and 98.13%, respectively. While our proposed GCN-MI had a higher accuracy (99.71%) this may be due to the consideration of the structure of leads. 

These studies used a limited number of subject records besides using many extracted beats from the same people. This situation can reduce the generalizability of the models, but our study used more than 10,000 people and each ECG record was obtained exclusively from one person. This set of data is available to the public, and one of its advantages is that our results are comparable to future studies on this database. None of the previous studies have considered the relationship of leads. Statistically, there is a type of linear or nonlinear correlation amongst heart leads, and calculating and using it for network training, as additional information, can be useful to categorize different types of heart rhythms. 

In our study, to use this additional information, we configured the GCN and calculated the relationship amongst cardiac leads using the MI matrix and provided it to the network as the adjacency matrix for the first time. Our network accuracy is high (99.71%) and this model has a good generalization ability for detecting ECG arrhythmias. Table 5 indicates the results of these studies.

## 5. Conclusions

Enhancing accuracy of the diagnosis tools for cardiac arrhythmias is very important to prevent the exacerbation of disease and possibly death. In recent years, the deep learning algorithms (with various network structures) have widely been used in the automatic detection of various arrhythmia types. In order to achieve high accuracy, constructing a network structure, which includes relationship between the ECG leads, plays an important role. This study provides a novel way to consider the structure between ECG data by constructing an efficient GCN model augmented with the adjacent matrix of MI. It was then illustrated that the constructed GCN-MI is very effective in diagnosing and classifying different types of arrhythmias and is significantly more accurate than the existing methods.

## Figures and Tables

**Figure 1 ijerph-19-10707-f001:**
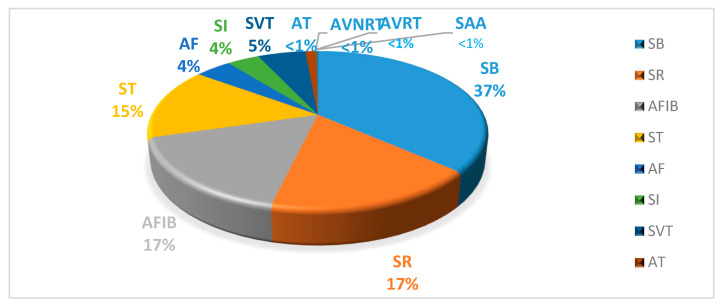
The distribution rate graph of rhythm classes across all records.

**Figure 2 ijerph-19-10707-f002:**
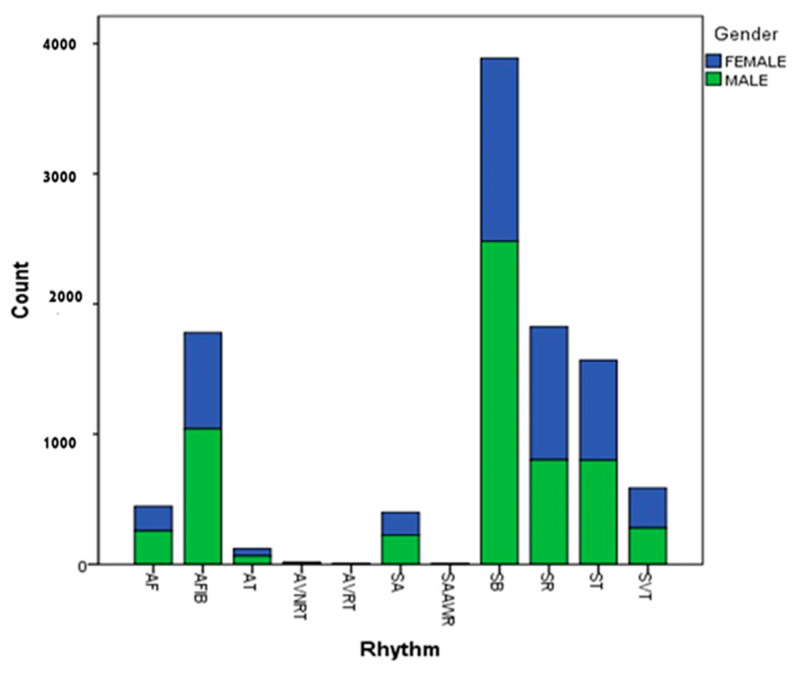
Frequency of rhythm by sex.

**Figure 3 ijerph-19-10707-f003:**
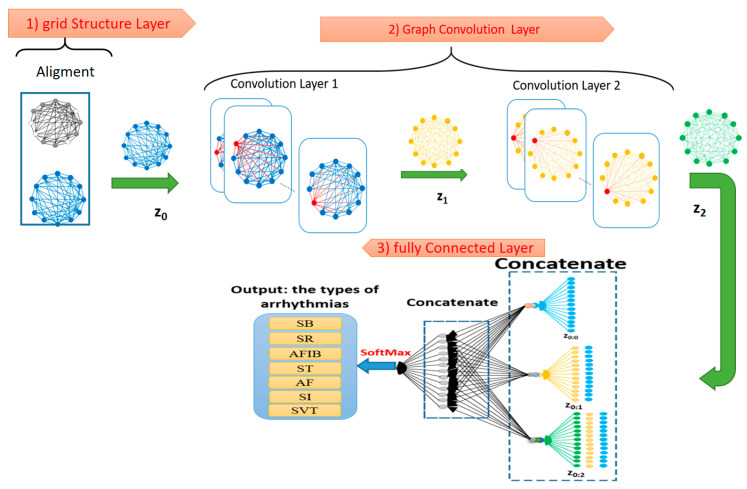
The network architecture for the GCN with two layers of convolution.

**Figure 4 ijerph-19-10707-f004:**
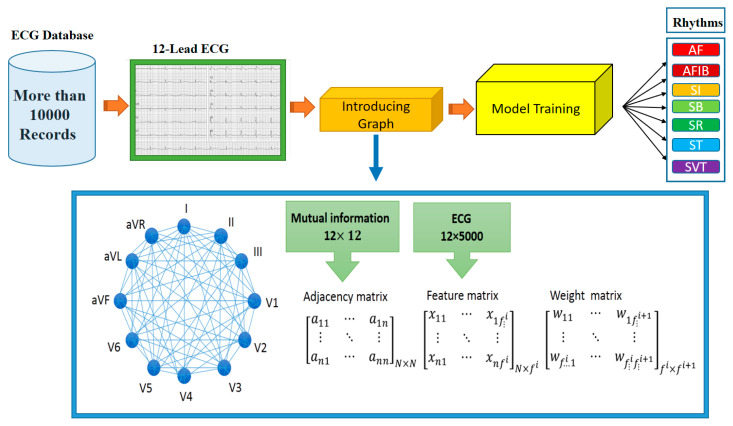
The proposed methodology and required materials for this study.

**Figure 5 ijerph-19-10707-f005:**
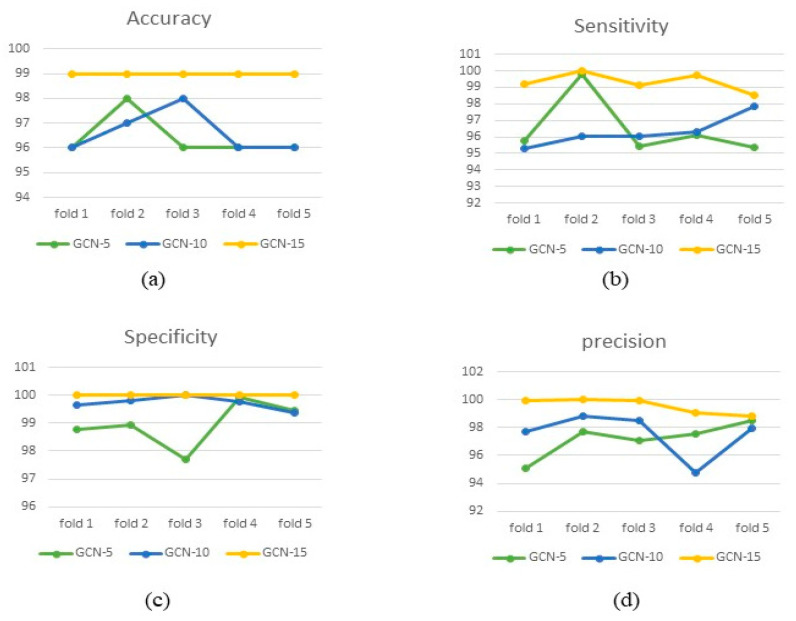
The performance values (including Accuracy, Sensitivity, Specificity and Precision) for GCN-MI-5, GCN-MI-10, and GCN-MI-15 using various k-fold CVs, are illustrated in subfigures (**a**–**d**), respectively.

**Figure 6 ijerph-19-10707-f006:**
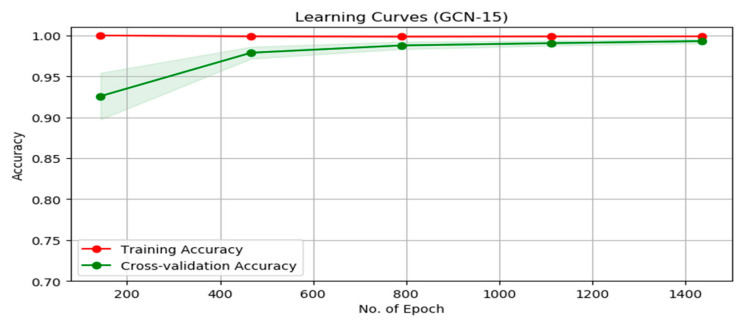
Learning curve for the GCN-MI with 15 layers.

**Figure 7 ijerph-19-10707-f007:**
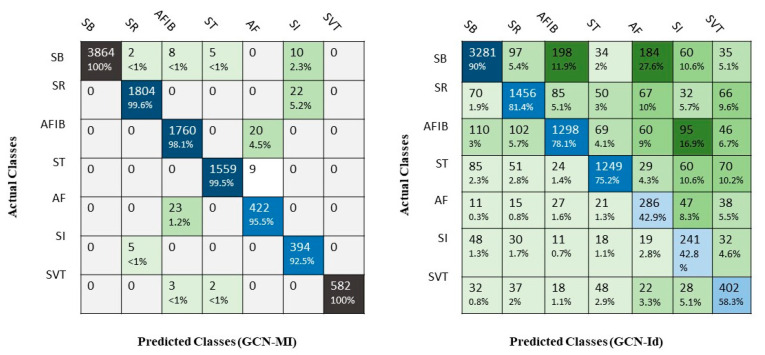
Confusion matrix for all records using GCN-MI and GCN-Id.

**Figure 8 ijerph-19-10707-f008:**
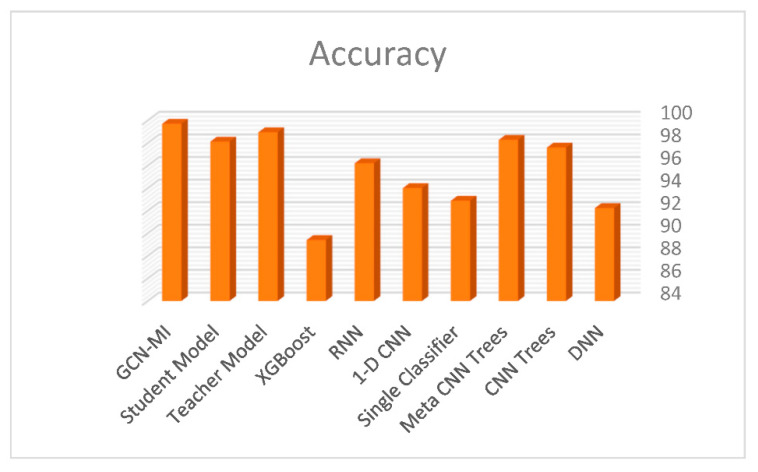
Comparison of deep learning models for arrhythmic diagnosis using the Chapman dataset.

**Table 1 ijerph-19-10707-t001:** Rhythm information and basic characteristics of the participants.

Acronym Name	Full Name	Frequency, n (%)	Age, Mean ± SD	Male, n (%)
SB	Sinus Bradycardia	3889 (36.53)	58.34 ± 13.95	2481 (58.48%)
SR	Sinus Rhythm	1826 (17.15)	54.35 ± 16.33	1024 (56.08%)
AFIB	Atrial Fibrillation	1780 (16.72)	73.36 ± 11.14	1041 (58.48%)
ST	Sinus Tachycardia	1568 (14.73)	54.57 ± 21.06	799 (50.96%)
AF	Atrial Flutter	445 (4.18)	71.07 ± 13.5	257 (57.75%)
SI	Sinus Irregularity	399 (3.75)	34.75 ± 23.03	223 (55.89%)
SVT	Supraventricular Tachycardia	587 (5.51)	55.62 ± 18.53	308 (52.47%)
AT	Atrial Tachycardia	121 (1.14)	65.72 ± 19.3	64 (52.89%)
AVNRT	Atrioventricular Node Reentrant Tachycardia	16 (0.15)	57.88 ± 17.34	12 (75%)
AVRT	Atrioventricular Reentrant Tachycardia	8 (0.07)	57.5 ± 16.84	5 (62.5%)
SAA	Sinus Atrium to Atrial Wandering Rhythm	7 (0.07)	51.14 ± 31.83	6 (85.71%)
All		10,646 (100)	51.19 ± 18.03	5956 (55.95%)

**Table 2 ijerph-19-10707-t002:** The performance values for GCN-MI-5, GCN-MI-10, and GCN-MI-15 in the testing set with the k-fold cross-validation method.

	Accuracy (%)			Sensitivity (%)			Specificity (%)			Precision (%)		
	GCN-MI			GCN-MI			GCN-MI			GCN-MI		
	5 Layer	10 Layer	15 Layer	5 Layer	10 Layer	15 Layer	5 Layer	10 Layer	15 Layer	5 Layer	10 Layer	15 Layer
Leave-one-out	96.57	96.82	99.39	95.76	95.32	99.22	98.78	99.66	100	95.08	97.68	99.94
k = 2	98.53	97.40	99.63	99.81	96.03	100	98.92	99.82	100	97.69	98.85	100
k = 3	96.93	98.56	99.83	95.40	96.06	99.15	97.70	100	100	97.09	98.50	99.98
k = 4	96.83	96.39	99.83	96.12	96.30	99.76	99.94	99.79	100	97.53	94.75	99.11
k = 5	96.96	96.63	99.24	95.34	97.87	98.52	99.46	99.39	100	95.55	97.97	98.87

**Table 3 ijerph-19-10707-t003:** The parameter values of the fitted GCN-MI network with 15 layers.

Parameter	Value
Learning Rate	0.02
Epochs	600
Hidden layers	15
Dropout	0.2
Weight Decay	10,000
Early Stopping	10

**Table 4 ijerph-19-10707-t004:** The performance values for all records separately for each class using GCN-MI and GCN-Id.

		Sensitivity	Precision	Specificity	Accuracy
SB	GCN-MI	99.35	100	100	99.76
	GCN-Id	90.21	84.36	89.02	89.49
SR	GCN-MI	98.79	99.61	99.92	99.72
	GCN-Id	81.43	79.73	94.80	92.12
AFIB	GCN-MI	98.88	98.10	99.61	99.48
	GCN-Id	78.14	72.92	93.48	90.67
ST	GCN-MI	99.43	99.55	99.92	99.85
	GCN-Id	83.88	79.65	95.62	93.62
AF	GCN-MI	94.83	95.47	99.80	99.59
	GCN-Id	42.88	64.27	98.03	93.83
SI	GCN-MI	98.75	92.49	99.68	99.65
	GCN-Id)	42.80	60.40	98.05	94.47
SVT	GCN-MI	99.15	100	100	99.95
	GCN-Id	58.34	68.48	97.68	94.56
Overall	GCN-MI	**98.45**	**97.89**	**99.85**	**99.71**
	GCN-Id	**68.24**	**72.83**	**95.24**	**92.68**

**Table 5 ijerph-19-10707-t005:** Performance comparison of the proposed method with other state-of-the-art methods.

Refs.	Study	Dataset	Num. of Subjects	Year	Method	Classes	Performance
[34]	Jiang et al.	PhysioNet/CinC Challenge 2020	512	2022	CNN+GCN	9	F-Score = 0.603
[35]	Shaker et al.	MIT-BIH	47	2020	GAN	15	Acc = 98.30%
[22]	Yao et al.	-	-	2020	ATI-CNN	8	Acc = 81.2%
[24]	Zhao & Tan	MIT-BIH	47	2020	CNN+ELM	4	Acc = 97.5%
[36]	Gao et al.	MIT-BIH	47	2019	LSTM, FL	8	Acc = 99.26%
[37]	Hannun et al.	-	53549	2019	DNN	12	AUC = 97%
[38]	Oh et al.	MIT-BIH	47	2019	Modified U-net	5	Acc = 97.32%
[39]	Li et al.	MIT-BIH	47	2019	ResNet	5	Acc = 99.38%
[40]	Yildirim et al.	MIT-BIH	47	2018	CNN	17	Acc = 91.33%
[41]	Xu et al.	MIT-BIH	47	2018	DNN	5	Acc = 93.1%
[42]	Acharya et al.	MIT-BIH	47	2017	CNN	5	Acc = 94.03%
[43]	Yildirim et al.	Chapman	10,588	2020	DNN	4	Acc = 96.13%
			10,436			7	Acc = 92.24%
[44]	Meqdad et al.	Chapman	10,646	2022	CNN Trees	7	Acc = 97.60%
[45]	Meqdad et al.	Chapman	10,646	2022	Meta CNN Trees	7	Acc = 98.29%
[46]	Mehari et al.	Chapman	10,646	2022	Single Classifier	7	Acc = 92.89%
[47]	Rahul et al.	Chapman	10,646	2022	1-D CNN	7	Acc = 94.01%
[48]	Kang et al.	Chapman	10,646	2022	RNN	7	Acc = 96.21%
[49]	Domazetoski et al.	Chapman	10,646	2022	XGBoost	**-**	Acc = 89.40%
[50]	Sepahvand et al.	Chapman	10,646	2022	Teacher model	**7**	Acc = 98.96%
					Student model	**7**	Acc = 98.13%
	**Proposed**	**Chapman**	**10,494**	**2022**	**GCN-MI**	**7**	**Acc = 99.71%**

## Data Availability

Data are available at https://figshare.com/collections/ChapmanECG/4560497/2 (accessed on 25 September 2020).

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
