# Peer review of "Developing Graph Convolutional Networks and Mutual Information for Arrhythmic Diagnosis Based on Multichannel ECG Signals"

_ijerph, 2022, doi:10.3390/ijerph191710707_

Round 1

Reviewer 1 Report

The manuscript was written and presented in high quality. It provided inside to the method and results. However, there is room for improve. My main concern is the abstract and discussion sections. at the first view, there is a gap between them, and need to reduced it. Also the format of abstract should modify based on the journal requirement.   

Author Response

We would like to thank Reviewer 1 for their positive evaluation of our manuscript and the feedback which have helped us to improve the manuscript. We agreed with your comments and revised the manuscript to address the comments accordingly.

  • The abstract was rewritten according to the journal format. Also, in rewriting the abstract, we have also eliminated the gap between the abstract and discussion sections. Furthermore, we have thoroughly proofread the whole paper.

Reviewer 2 Report

1st Criterium: Clarity of the presentation and the quality of English usage:

1. "Background: " should be eliminated from the Abstract.

2. "Cardiovascular diseases like arrhythmias" - when you use the generic name of an illness or health condition, use it as a singular, in this case, "arrhythmia". Arrhythmia is the generic name of this health condition. "An arrhythmia" is an event, an episode during which the medical condition becomes obvious by the fact that the heart could beat too slow, too fast, or purely irregular. "Arrhythmias" - is a series of such events. 

3. "Cardiac arrhythmia [...] is an initial category of cardiovascular disorders [4]" - this doesn't make much sense

4. "the 12-lead-ECG is still used as the "true standard" for assessing a wide range of heart diseases after 100 years" - a better formulation is necessary

5. "Arrhythmia-type is realized by the ECG waveform, which accuracy and velocity of diagnosis play a significant role in both preventing and treating heart diseases" - needs to be revisited/clarified/rephrased. As it is now, it looks trivial and out of English. 

6. "clinical-decisions" I cannot see any reason why hyphenation is necessary here.

7. "human-errors", "machine-learning schemes" , "noise-cancellation", "data-normalization" Idem 6

8. "Recently, using neural networks and deep learning (DL) are highly applied to analyze" - unusual English usage (again)!

9. "One type of neural network, that alleviates these problems somewhat" - - unusual English usage (again)!

10. "A convolutional neural network is a group of deep neural networks generally used for visual image analysis [16], which also work well for audio inputs and signals." - a shallow view on the CNN topic

11. "GCNs are a group of deep neural net-142 works that perform mathematical operations called convulsions on graphs" - convulsions ?!

C1: the clarity of the presentation and the quality of English usage are definitely not the strong points of this manuscript.  

2nd Criterium: The clarity of mathematical presentation and reasoning:

1. Paranthesation is not declared on line 198

2. Formulas on lines 196 and 198 are not equivalent

3. The definition of MI is formulated in a way suggesting that it can be accurately/equivalently translated in terms of conditional probabilities as stated on line 198. That is not true. 

4. "For the continuous state, all relations with differential can be also expressed (see [31-33] for further details). This study calculated the ??_{?×?} matrix for 12-ECG-leads by R software." - Doubtable at least. Not clear enough. Who is N? For continuous probabilities (not states! what is a state here? where is this concept defined before being used) sums should be translated to integrals, not to differentials!

5. is not clear how calculable is H on line 157

C2: Mathematical reasoning and presentation could be much better. 

3rd Criterium: Dataset understanding and Results reporting:

As a fact:

Accuracy = (sensitivity) (prevalence) + (specificity) (1 - prevalence).

where prevalence is the probability of a given disease/condition in the population at a given time, which in this case is known because the dataset is given. 

1. The above relation cannot be validated on data reported within Table 2

C3: Results reporting technique is not convincing enough. Reported numerical values cannot pass cross-validation with a priori data known from the dataset. 

4th Criterium: Novelty

1. the paper "Diagnostic of Multiple Cardiac Disorders from 12-lead ECGs Using Graph Convolutional Network Based Multi-label Classification" already dealt with the same problem, but only in a much better manner than the current manuscript. 

C4: The novelty of the submitted manuscript is clearly questionable

Reviewer 3 Report

The main concept of the paper is innovative and the results confirm the quality of the models' structure. It needs more extensive analysis, on the comparison sub-section, before the discussion. Also, please, describe the architecture of the main model, parameterized, with emphasis on the weights and how the information is transferred through the nodes. 

Author Response

We would like to thank Reviewer 3 for their time in providing detailed and highly constructive feedback on our paper. Further, we would like to thank them for their positive evaluation of our study, acknowledging the strengths of the work and its appropriateness for the IJERPH journal.

We agreed with your comments and revised the manuscript to address the comments accordingly. The details of changes made in the revised paper are:

  • The architecture of the main selected network and its parameters were discussed in Section 3, please see lines 285-294 and Table 3 in Section 3.
  • We have also further conducted the t-test to compare the accuracy of GCN-MI and GCN-Id models, and the derived results were discussed in lines 330-334, just after Table 4.
  • We have also entirely proofread the whole paper.

Reviewer 4 Report

Thanks authors to present this work, which propose a GCN-MI to classify arrhythmic based on ECG dataset. Although lots of work on it, this paper give a clear description on network design/analysis/discussion. I would like to recommend the paper for publication. Just one comment for authors:

   I don't think the three GCN network  structure with different layers-5, 10, 15 is a key creative point. It's obvious that more layer more accuracy more time consumption, ..., no free lunch.

  Besides, in abstract, I would like to suggest to implement your innovation aspect, as list in Lines 105-110. I guess the GCN-MI, which maybe more promising for readers. 

Author Response

We would like to thank Reviewer 4 for their time in providing detailed and highly constructive feedback on our paper. Further, we would like to thank them for their positive evaluation of our study, acknowledging the strengths of the work and its appropriateness for the IJERPH journal.

We agreed with your comments and revised the manuscript to address the comments accordingly. Please see our responses to your comments below:

Just one comment for authors:

 Comment 4.1: I don't think the three GCN network structure with different layers-5, 10, 15 is a key creative point. It's obvious that more layer more accuracy more time consumption, ..., no free lunch.

Response 4.1: Many thanks for this comment. We agree with your comment. However, the main reason of trying the GCN model with various layers was to select the optimal model (i.e., simplest model with the desired accuracy level) with the acceptable computational cost for training the network. We thus added the following discussion in lines 284-290 to make it clear that why there would be a limit to construct a deep model:

However, it is obvious that adding more layers will help us to extract more features, but we should be aware of the possible overfitting by adding more layers and computational layers. Based on the above benchmark study between the accuracy and other predictive performance values of the GCN-MI with different layers, and negligible increased computational cost, the GCN-MI with 15 intermediate layers was thus selected and configured by cross-validation method and with fold=4.

Comment 4.2:  Besides, in abstract, I would like to suggest implementing your innovation aspect, as list in Lines 105-110. I guess the GCN-MI, which maybe more promising for readers. 

Response 4.2: Many thanks for this comment. The abstract of the study was rewritten. In the abstract, instead of emphasizing the importance of comparing three GCN networks with different layers-5, 10, 15, we tried to focus on the innovation of GCN-MI design and use.

Round 2

Reviewer 2 Report

The authors considered the review remarks previously given to them. The manuscript can be published in the current form. 

Author Response

We would like to thank Reviewer 2 for their time in checking our responses and suggesting our paper for publication in the IJERPH journal.

We have thoroughly gone through the paper, checked “English language and style” and corrected any mistakes we found. Furthermore, we have fixed several minor spelling typos.

Once again, we would like to thank Reviewer 2 for their positive evaluation of our study, acknowledging the strengths of the work and its appropriateness for the IJERPH journal.

Reviewer 3 Report

I find the content of this article, after an analytical examination, very important and innovative. The models are already constitute the state of the art, on the field of graph convolutional networks, but the specific implementation on this dataset has significant impact on the medical section. After a minor revision, by providing a more extensive analysis on the architectures and giving more figures that show the pre-processing steps, the paper could be a valuable contribution in the scientific community.

Author Response

We would like to thank Reviewer 3 for their time in providing detailed and highly constructive feedback on our paper. Further, we would like to thank them for their positive evaluation of our study, acknowledging the strengths of the work and its appropriateness for the IJERPH journal.

Regarding the pre-processing steps, we should say that the ECG data used in our paper was recently collected and published in Zheng et al. (2020) (see Reference [5]). In addition, they conducted pre-processing steps on the raw data to clean the data and remove the noise from the raw ECG data. We have added a discussion in “Section 2.4”, lines 241-267, and explained steps required to construct the proposed method and classifier, including data pre-processing step.   

We have thoroughly gone through the paper, checked “English language and style” and corrected any mistakes we found. Furthermore, we have fixed several minor spelling typos.

Once again, we would like to thank Reviewer 3 for their positive evaluation of our study, acknowledging the strengths of the work and its appropriateness for the IJERPH journal.